

# Consistency of dimensional distributions and refractive indices of desert dust measured over Lampedusa with IASI radiances

Giuliano Liuzzi[1], Guido Masiello[1], Carmine Serio[1], Daniela Meloni[2], Claudia Di Biagio[3], and Paola Formenti[3]

[1]School of Engineering, University of Basilicata, Via dell'Ateneo Lucano 10, Potenza, Italy
[2]ENEA, Laboratory for Observations and Analyses of the Earth and Climate, Via Anguillarese 301, Rome, Italy
[3]Laboratoire Interuniversitaire des Systémes Atmosphériques (LISA), UMR-CNRS 7583, Créteil, France

*Correspondence to:* Giuliano Liuzzi
giuliano.liuzzi@unibas.it

**Abstract.** In the context of the ChArMEx campaign, we present here some results concerning the quantitative comparison between simulated and observed radiances during a dust event occurred between June and July 2013 in the southern Mediterranean basin, involving the airmass above Lampedusa island. In particular, comparisons have been performed between radiances as observed by the Infrared Atmospheric Sounder Interferometer (IASI) and those simulated using the $\sigma$-IASI-as

radiative transfer model, which takes into account aerosol extinction effect through a set of fast parameterizations. Simulations have been carried on with different sets of input complex refractive indices, which take into account the parent soils of the aerosols, and using the high-quality characterization of desert dust aerosol microphysical properties, achieved through direct measurements in the ChArMEx experiment; on the one hand, this comparison has offered the possibility to test the feasibility of the radiative transfer model. On the other hand, this work goes through a direct validation of different refractive indices sets

for desert dust in the thermal infrared. Results show a good consistency between calculations and observations, especially in the spectral interval 800-1000 cm$^{-1}$; moreover, the comparison between calculations and observations suggests that further efforts are needed to better characterize desert dust optical properties in the short wave (above 2000 cm$^{-1}$). In any case, we show that it is necessary to properly tune the refractive indices according to the geographical origin of the observed aerosol.

## 1   Introduction

The discussion about the possibility to detect and characterize atmospheric aerosols in the thermal infrared spectral range is relatively recent (Hollweg et al. , 2006; Clarisse et al. , 2013) and very active, since infrared satellite observations enable an extensive spatial-temporal characterization of aerosol optical and microphysical properties, and to quantitatively describe their transport processes (e.g. Peyridieu et al. (2013); Sellitto et al. (2016); Prata and Prata (2012)). However, a proficient exploitation of satellite infrared data requires a good knowledge of the spectral properties of the most common atmospheric

aerosols in this spectral range.

In this context, there could be several sources of uncertainty, some of them being: a poor characterization of complex refractive indices, the particle size distribution, the aerosol's vertical profile on the spatial scale of interest (Vandenbussche et al. , 2013),



and the degree of robustness of radiative transfer, which should be capable to consistently reproduce the radiative impact of aerosols.

All these issues have been subjected to investigations, which led to a significant improvement of the physics of the radiative transfer in presence of aerosols, as far as both accuracy and computational performances are concerned. The two aspects

have the same importance, since the quality and the amount of infrared data produced by satellite-based sensors have both steeply increased in the last decade. In this respect, before going through the exploration of aerosol properties, we describe the radiative transfer model we use in this work, which is called $\sigma$-IASI-as, focusing on the scheme used by the code to simulate the radiative effect of clouds and aerosols, and on its computational performances. The model, which is conceived mainly for research purposes, has been already successfully used in the simultaneous retrieval of several polluting gaseous species

together with the thermodynamic state of surface and atmosphere (Liuzzi et al. , 2016) with the Infrared Atmospheric Sounder Interferometer (IASI, Hilton et al. (2012)). In this work, instead, we use the model for simulating the observed radiance in presence of aerosols and clouds. In this sense, the new model constitutes an advancement with respect to the former $\sigma$-IASI model (Amato et al. , 2002). In this respect, it is fair to point out that $\sigma$-IASI-as computes the extinction due to aerosol/cloud particles adopting a strategy which is similar to that of other fast radiative transfer models, such as the Radiative Transfer for

TOVs model (RTTOV, Saunders et al. (2013)), massively employed in data assimilation in the context of Numerical Weather Prediction (NWP), an approach which is largely faster than the exact multiple scattering effects calculation implemented in line-by-line models such as LBLRTM (Clough et al. , 2005), and the Discrete Ordinates Radiative Transfer Program for a Multi-Layered Plane-Parallel Medium (DisORT, Stamnes et al. (1988)). The scheme used in $\sigma$-IASI-as is a compromise between a fully parameterized one, and an exact multiple scattering algorithm, in order to retain, at the same time, an high

degree of generality and the maximum fastness. Furthermore, like its predecessor $\sigma$-IASI, $\sigma$-IASI-as is not customized for any particular instrument, and works with a pseudo-monochromatic approach: in this way the model is of more general application than the majority of predictors-based radiative transfer models.

The present study completes the picture of the model capabilities, which have been already investigated in depth in Liuzzi et al. (2014, 2016) as far as gases are concerned. As stated before, an incomplete or inaccurate characterization of the atmospheric

aerosol could be the greatest source of uncertainty in radiative transfer calculations. Thus, for this validation exercise, we have fruitfully employed the aerosol microphysical properties (i.e. dimensional distribution and concentration) derived during the Chemistry-Aerosol Mediterranean Experiment (ChArMEx, Mallet et al. (2016)) Special Observation Period SOP1a concurrent with the Aerosol Direct Radiative Impact on the regional climate in the MEDiteranean region (ADRIMED) field campaign by the measurements made on board the ATR-42 over Lampedusa, in four different days between June and July 2013. Indeed,

natural aerosols largely affect the radiative transfer in the Mediterranean basin atmosphere in a direct or indirect way (e.g. Di Sarra et al. , 2008, 2011; Pace et al. , 2005, 2006; Di Biagio et al. , 2010; Meloni et al. , 2015). At the location and the time of the observations we have chosen, the dominant contribution has been found to be that of the desert dust emissions from various Saharan soils. Indeed, for our purposes, we have used the dimensional properties and profiles derived from ATR-42 flights (Denjean et al. , 2016), and produced radiative transfer calculations with different sets of complex refractive indices,

according to the soil from which aerosol have been observed to come: such an exercise is devoted to quantify the accuracy of



our knowledge of desert dust refractive index in the thermal infrared. This will be made on the full IASI spectral coverage (3.62 to 15.5 $\mu$m) in order to see whether or not input refractive indices fail to represent the observed radiance in spectral regions dominated either from aerosol absorption or scattering.

The paper is organized as follows: in Section 2 we will describe the $\sigma$-IASI-as model, with the aerosol extinction calculation scheme implemented therein; Section 3 is dedicated to show the IASI data and related ancillary information used in this case study. In Section 4 we will conduct an in-depth analysis of results, as far as both radiative transfer scheme consistency and aerosol spectral properties are concerned. Conclusions will be drawn in the end.

## 2 The $\sigma$-IASI-as radiative transfer model

The radiative transfer model we work with is called $\sigma$-IASI-as. As already stated before, it is an advanced version of the $\sigma$-IASI model (Amato et al. , 2002) as far as clouds and aerosols treatment is concerned. The model is tailored to compute the Earth/atmosphere-emitted radiance in the wave number interval 100-3000 cm$^{-1}$, and works within the assumption of a plane-parallel geometry. Moreover, even if most of the forthcoming discussion is developed in the hypothesis of an observer located at the top of atmosphere, the model is capable to compute both ground-based and satellite-based radiance.

The radiative transfer equation solved in $\sigma$-IASI-as takes the subsequent form, in which the total observed radiance $R(\sigma)$ at wave number $\sigma$ is expressed as the sum of four terms:

$$
\begin{aligned}
R(\sigma) = R^{surf}(\sigma) + R^{\uparrow}(\sigma) + R^{\downarrow}(\sigma) + R^{sun}(\sigma) = \\
\epsilon_g B(T_g)\tau_0 + \int\limits_{0}^{+\infty} B(T)\frac{\partial\tau}{\partial z}\mathrm{d}z + \\
+ (\epsilon_g - 1)\tau_0 \int\limits_{0}^{+\infty} B(T)\frac{\partial\tau^f_*}{\partial z}\mathrm{d}z + \frac{1-\epsilon_g}{\pi}\tau'\mu_s I_s
\end{aligned} \tag{1}
$$

where, respectively: the first term represents the radiance emitted directly by the surface, with a skin temperature $T_g$, an emissivity $\epsilon_g(\sigma)$, and attenuated by the atmosphere along the observation path, whose total transmittance is $\tau_0(\sigma)$; the second term is the up-welling radiance, namely the radiance emitted directly from the atmosphere along the slant path in the viewing direction, integrated along the path itself; the third term expresses the down-welling radiance, which is the radiance contribution emitted from the atmosphere towards surface, back-reflected from it and that reaches the observer; according to the type of surface, the term $\tau^f_*$ is computed in different ways; if the surface is a Lambertian diffuser, $\tau^f_*$ is the diffuse transmittance, that can be calculated as the transmittance function at a suitable angle, corresponding to an effective zenith angle $\theta_r$=52.96 degrees (Elsasser , 1942). In the case of a specular reflector (e.g. sea surface), instead, $\tau^f_*$ will be the nadir transmittance, exponentially rescaled by the cosine of the viewing zenith angle (VZA). Finally, the last term represents the solar radiance $I_s/\pi$ reflected back from surface and then reaching the observer, with $\tau'$ the two-way transmittance along the path Sun-surface-observer, and $\mu_s$ the cosine of the solar zenith angle. The solar irradiance $I_s$ implemented in the model is borrowed by the Kurucz model (Chance and Kurucz , 2010). Formally speaking, $\tau'$ is the only term that undergoes a modification, according to the geometry





(nadir-looking or zenith-looking): in this second case, the two-way transmittance simplifies to one-way, along the observation path. The other terms remain unaltered.

In a non-scattering atmosphere, eq. (1) can be used to estimate the observed radiance both in clear sky and in presence of aerosol/cloud particles. Under the further limitation that the instrumental Field of View (FOV) is not affected by cloud shadows

or other weird inhomogeneities, the total observed radiance will be the sum of two terms, one of clear sky and the other that is referred to the overcast fraction $\alpha$ of the FOV (Chanine , 1974):

$$R(\sigma) = (1-\alpha)R_C(\sigma) + \alpha R_N(\sigma) \tag{2}$$

The term $R_C(\sigma)$ in Eq. (2) is explicited by Eq. (1), and is the clear sky radiance. The cloudy radiance $R_N(\sigma)$, at a first order, is formally calculated in the same way; it goes without saying that, instead, the transmittance will be calculated taking into

10 account not only the gases optical depths, but also clouds opacity. At a second order, it ought be considered an additional term that accounts for possible sunglint effects caused by reflection of solar radiation by clouds top or sea surface. Such contribution becomes important only at wave numbers greater than 2000 cm$^{-1}$, while it is negligible in the thermal infrared; however, since in the analysis we will conduct we preliminarily discard spectra affected by such effects, the calculation of this term is not implemented in the $\sigma$-IASI-as model.

The radiative transfer equation (1) is implemented in the model solving it by dividing the atmosphere in a certain number of discrete layers, each one assumed to be characterized by an homogeneous temperature and composition. Hence, one can write two expressions for $R_C$ and $R_N$:

$$R_C = \epsilon_g B(T_g)\tau_{0,C} + \sum_{j=1}^{N_L} B(T_j)(\tau_{C,j} - \tau_{C,j-1}) +$$
$$+ (\epsilon_g - 1)\tau_{0,C} \sum_{j=1}^{N_L} B(T_j)(\tau^f_{C,j*} - \tau^f_{C,j-1*}) + \frac{1-\epsilon_g}{\pi}\tau'_C \mu_s I_s \tag{3}$$

$$R_N = \epsilon_g B(T_g)\tau_{0,N} + \sum_{j=1}^{N_L} B(T_j)(\tau_{N,j} - \tau_{N,j-1}) +$$

$$+ (\epsilon_g - 1)\tau_{0,N} \sum_{j=1}^{N_L} B(T_j)(\tau^f_{N,j*} - \tau^f_{N,j-1*}) + \frac{1-\epsilon_g}{\pi}\tau'_N \mu_s I_s \tag{4}$$

The dependence on the wave number $\sigma$ is omitted for brevity. In both equations, $\tau_j$ is the total transmittance from the top of the $j$-th layer to $\infty$, and $\tau_{j-1}$ is the total transmittance from the bottom of the $j$-th layer to $\infty$, leading $\tau_{N_L}$=1. Clarifications are provided within Fig. 1. In addition, $T_j$ is the equivalent average temperature of the bulk of the $j$-th layer. In $\sigma$-IASI-as, the atmospheric grid is made by 60 layers, whose pressure boundaries are fixed, and span the interval 1013-0.005 mbar: hence,

the number of layers of which transmittance is computed depends strictly on surface pressure, taking into account the airmass above the observed target.

Transmittances in the clear-sky and cloudy radiances are defined according to the considered absorbers. In $\sigma$-IASI-as it is



assumed that the clear-sky transmittance is affected both by gases and aerosol particles, namely assuming that aerosols are uniformly distributed on the entire FOV; the extinction due to water and ice clouds, together with that due to gases and other aerosols, yield the transmittance in the cloudy portion $\alpha$ of the FOV. Looking at Figure 1, it is possible to write explicitly the expressions for the transmittance $\tau_j$ in the cases of clear and cloudy sky:

$$\tau_{C,j} = \prod_{i=1}^{j} \exp(-\chi_{C,i}) = \prod_{i=1}^{j} \exp\left[-(\chi_{gas,i} + \chi_{aer,i})\right] \tag{5}$$

$$\tau_{N,j} = \tau_{C,j} \cdot \prod_{i=1}^{j} \exp\left(-\chi_{cld,i}\right) \tag{6}$$

where $\chi_{C,i}$ and $\chi_{N,i}$ are the single layer total optical depths in absence/presence of clouds, and $\chi_{aer,i}$ is the optical depth of aerosol particles in the $i$-th layer.

Calling $\eta_j$ the single-layer transmittance of the $j$-th layer, and using the definitions provided so far, we can better express the relation between $\eta_j$ and the sum of the optical depths of all the atmospheric absorbers in the layer itself:

$$\eta_j(\sigma) = \frac{\tau_{j-1}(\sigma)}{\tau_j(\sigma)} = \exp\left[-\sum_{i=1}^{S} \chi_{i,j}(\sigma)\right] \tag{7}$$

being $S$ the number of atmospheric absorbers, and $\chi_{i,j}(\sigma)$ the monochromatic optical depth of the $i$-th species from the top of the $j$-th layer to its bottom at wave number $\sigma$. The simplest way to make explicit $\chi_{i,j}(\sigma)$ is to recall the well known Lambert-Beer's law:

$$\chi_{i,j}(\sigma) = q_{i,j} k_{i,j}(\sigma) \Delta h_j \tag{8}$$

where $q_{i,j}$ is the concentration of the $i$-th species in the $j$-th layer, $k_{i,j}(\sigma)$ is the absorption coefficient for that species at wave number $\sigma$, and $\Delta h_j = H_j / \cos\theta$ denotes the path length, along the $\theta$-direction of observation, from the bottom of the $j$-th layer to its top, being $H_j$ the thickness of the $j$-th layer.

The way in which the term $k_{i,j}(\sigma)$ is treated in the model depends strictly on the type of absorber: since a different physics regulates the interaction of radiation with gases and particles, the estimation of $k_{i,j}(\sigma)$ relies on two distinct approaches, one for gases and one for aerosols and clouds, which are described here in the following.

## 2.1 Gases optical depths

The $\sigma$-IASI-as code calculation of gases' optical depths is based on a scheme which leaves aside the direct manipulation of spectroscopy. On the contrary, the approach followed is that of working in a pseudo-monochromatic context, in which transmittances are calculated on an equally-spaced wave number grid. To do this, the $\sigma$-IASI-as architecture embodies a wide look-up table where, for each layer, atmospheric species and wave numbers, optical depths are pre-computed and stored. Then, optical depths are rescaled with air pressure and temperature: on the one hand, pressure is fixed by the different atmospheric





layers; on the other hand, the dependence on temperature is parameterized: purely monochromatic optical depths are generated using the version 12.2 of LBLRTM (Clough et al. , 2005), equipped with the spectral library AER v_3.2 with the continuum model MT-CKD v_2.5.2 (Mlawer et al. , 2012) using as input parameters the reference temperature profile U.S. Standard Atmosphere 1962 (Anderson et al. , 1986), with the associated reference gases concentrations. Without varying them, optical depths are re-calculated using the same temperature profile translated by eight evenly spaced temperature values going from -40 K to + 40 K. This choice is adequate to take into account the usual temperature variations that occur in the Earth's atmosphere. Once optical depths are calculated, for each individual species (except for water vapour), layer and wave number, the behaviour of the optical depth with temperature is parameterized by a second order polynomial:

$$\chi^{\sigma}_{i,j} = q_{i,j} \left( c^{\sigma}_{0,i,j} + c^{\sigma}_{1,i,j} \Delta T_j + c^{\sigma}_{2,i,j} (\Delta T_j)^2 \right) \tag{9}$$

where $\Delta T_j$ is the difference between the reference temperature profile and the actual equivalent temperature of the $j$-th layer, $q_{i,j}$ is the concentration of the $i$-th gas in the $j$-th layer. Water vapour optical depth, instead, is modelled using a slightly different parameterization: in order to take into account the effects depending on gas concentration, such as self-broadening of spectral lines, the polynomial expression includes a further coefficient $c^{\sigma}_{3,1,j}$ (Masiello and Serio , 2003), which multiplies the water vapour concentration:

$$\chi^{\sigma}_{1,j} = q_{1,j} \left( c^{\sigma}_{0,1,j} + c^{\sigma}_{1,1,j} \Delta T_j + c^{\sigma}_{2,1,j} (\Delta T_j)^2 + c^{\sigma}_{3,1,j} q_{1,j} \right) \tag{10}$$

where the water vapour species is denoted by $i = 1$, holding the species ordering of the HITRAN database (Rothman et al. , 2013). In order to reduce the dimensionality of calculations with respect to line-by-line models, the coefficients of the polynomial parameterizations are binned on a pseudo- monochromatic spectral grid with $\Delta\sigma = 10^{-2}$ cm$^{-1}$.

The present version of $\sigma$-IASI-as can compute the observed radiance with an user-specified temperature and emissivity, atmospheric profiles of temperature, and specific profiles for $H_2O$, $HDO$, $O_3$, $CO_2$, $CO$, $CH_4$, $N_2O$, $HNO_3$, $SO_2$, $NH_3$, $OCS$ and $CF_4$. Besides these gases, the code computes the optical depth of an ensemble of other gases whose concentration is kept fixed (mixed gases). This set includes the major species $N_2$ and $O_2$, which are considered through their continuum. Moreover, the model considers other trace gases, such as $NO$, $NO_2$, $HCl$, $HCN$, $CH_3Cl$, $OH$, $H_2CO$ and $C_2H_2$. Their reference vertical profiles are fixed according to the U.S. Standard atmosphere (Anderson et al. , 1986). Mixed gases include also the heavy CFCs molecules $CCl_4$, CFC-11, CFC-12 and HCFC-22. Their column abundance is scaled consistently with the most recent World Data Center for Greenhouse Gases report (WDCGG , 2015). It is fair to point out that, among the available fast radiative transfer models, $\sigma$-IASI-as is the more flexible in terms of number of gaseous species whose concentration can be tuned. Furthermore, the analytical parameterization expressed by Eq. (9-10) enables the possibility to compute, within the model, the derivatives of the radiance (namely, Jacobians) with respect to all the atmospheric and surface parameters mentioned above (for details see Amato et al. (2002)).

## 2.2 Aerosols and clouds optical depth

The $\sigma$-IASI-as model works with a physically-based method also to compute the extinction due to aerosol particles and clouds, which are based essentially on the same physics. To treat them, the model exploits an ab-initio approach: the code integrates





Mie routines (Bohren and Huffmann , 2000; Massie and Hervig , 2013) which are called iteratively within the calculation of single-layer transmittances. The results of Mie calculations are manipulated according to the scheme described in (Chou et al. , 1999) for reckoning an effective aerosols and clouds optical depth. According to this, the aerosol optical depth at a given wave number $\sigma$ and a given layer $j$ is computed as follows:

$$\chi_{aer,j}(\sigma) = q_{aer,j} k_{aer,j}(\sigma) \Delta h_j \tag{11}$$

where $q_{aer,j}$ is the aerosol concentration [particles cm$^{-3}$], $\Delta h_j$ has the same meaning as in Eq. (8), while $k_{aer,j}(\sigma)$ is the equivalent aerosol extinction per particle [km$^{-1}$].

In the scheme by Chou et al. , $k_{aer,j}(\sigma)$ includes the effects of three processes: emission, absorption and scattering. Actually these three processes are quite coupled, and there is no way to exactly take them into account at the same time. Despite this, there are several possible approximations that can be used to do this: the scheme implemented in $\sigma$-IASI-as embodies an ad-hoc parameterization which estimates multiple scattering effects in the long waves, yielding an equivalent aerosol optical depth. In this sense, this is comparable to an "absorption approach".

Hence, at a given wave number $\sigma$ and for the layer $j$-th (subsequently omitted for conciseness), the term $k_{aer,j}(\sigma)$ can be expressed as:

$$k_{aer}(\sigma) = \beta_{ext}(\sigma) \big[ (1 - \omega(\sigma)) + b(\sigma)\omega(\sigma) \big] \tag{12}$$

where $\beta_{ext}(\sigma)$ is the extinction efficiency per particle [km$^{-1}$], $\omega(\sigma)$ is the single-scattering albedo, and $b(\sigma)$ is the mean fraction of radiation scattered in the upward direction. In the hypothesis that the incoming radiation is isotropic, $b(\sigma)$ is the integral average in the upward direction (first integral) of the cumulative function (second integral) of the scattering phase function:

$$b(\sigma) = \frac{1}{2} \int\limits_0^1 \mathrm{d}\mu \int\limits_{-1}^0 \mathrm{d}\mu' P(\mu, \mu', \sigma) \tag{13}$$

where $\mu = \cos(\theta)$ and $P(\mu, \mu', \sigma)$ is indeed the scattering phase function. In this way, the code avoids the exact calculation of multiple scattering effects, by parameterizing them in a semi-analytical way. For the phase function, the code implements the most common one for atmospheric applications, namely the Henyey-Greenstein function (Henyey & Greenstein , 1941) which, again, is not directly computed by its original integral form, in order not to introduce an unsustainable computational load. Instead, $b(\sigma)$ is computed as a third-order polynomial of the asymmetry parameter $g(\sigma)$:

$$b(\sigma) = 1 - \sum_{l=1}^4 a_l g^{l-1}(\sigma) \tag{14}$$

where the four coefficients of this linear combination are $a_1$=0.5, $a_2$=0.3738, $a_3$=0.0076, and $a_4$=0.1186.

All the quantities involved in the aerosol/cloud extinction calculations depend on the microphysical properties of particles, namely their dimensional distribution and their complex refractive indices. The $\sigma$-IASI-as can compute aerosol and clouds





extinction due to an arbitrary number of aerosols and a superposition of an used-defined number of log-normal modes, which are computed within the code on the basis of user-defined average radius, standard deviation, cut-offs and concentration on 100 points. Calculations are performed on a wave number grid with $\Delta\sigma=15$ cm$^{-1}$, which is fine enough to correctly reproduce the absorption features of all the atmospheric aerosols. Results are subsequently interpolated on the wave number grid used to

compute gases optical depths.

$\sigma$-IASI-as code has a built-in routine that embodies the complex refractive indices of the most common atmospheric aerosols, borrowed from the existing databases (Shettle and Fenn , 1979; Hess et al. , 1998; Massie and Hervig , 2013), and of water ice (Warren , 1984), and water vapour for clouds extinction calculation. The aerosols handled by the code are listed in Table 1. For these aerosols, the code can compute also radiance Jacobians with respect to their concentration.

## 2.3   Code performances and potentialities

The computational performances of the code vary according to the complexity of clouds and aerosols vertical profiles, the number of log-normal modes involved, and to the fact that Jacobians are computed or not. A clear-sky single IASI spectrum (8461 channels) needs 1.0 sec to be computed on a modern CPU. This time increases by $\sim$0.1 sec for each log-normal aerosol/cloud mode; this is largely comparable to other fast models such as RTTOV (Faijan et al. , 2012). The calculation of a full IASI

spectrum with all the Jacobians (with respect to surface temperature, emissivity, temperature profile, gas and aerosol/cloud concentrations) requires some $\sim$6.0 sec, that means 0.04 sec per 50 spectral channels. The time is comprehensive of that required to read the optical depths look-up table.

As already stated, the way in which the code computes the observed radiance is not dependent on the instrumental technical characteristics, because of the pseudo-monochromatic approach, differently from other, common parametric methods based on

predictors. This feature makes the code adaptable to any kind of instrument which observes in the infrared, simply by changing the spectral response function in the code, and not its architecture, which is an unique in the context of fast radiative transfer models.

Another relevant features of $\sigma$-IASI-as is its capability to deal with different surface types, hence different reflection geometries from surface, both Lambertian (e.g. on most of land surfaces) and specular (e.g. on the sea surface), without any degradation

in the code performances.

Moreover, the inclusion of a large variety of atmospheric aerosols in the code, allows to use the code both in an operational context, and for pure research applications, like in the case we are going to show here subsequently.

## 3   Data and methods

### 3.1   IASI data

The validation accomplished here consists of a direct comparison between the radiance simulated using $\sigma$-IASI-as, which requires atmospheric profiles of temperature, gases, and aerosols, together with their spectral refractive indices, and their





dimensional characterization, and the collocated observations in the thermal infrared from the IASI interferometer.

IASI (Hilton et al. , 2012) has been developed in France by CNES and is flying on board the MetOp platforms. The two MetOp satellites (A and B) are part of the EUMETSAT European Polar System (EPS). IASI has been conceived for meteorological studies; hence, its main aim is to provide suitable information on the thermodynamic status of atmosphere (temperature and

water vapour profiles) and surface. The instrument works in the whole spectral interval from 645 to 2760 cm$^{-1}$, with a sampling interval $\Delta\sigma$ equal to 0.25 cm$^{-1}$ and an effective apodized resolution of 0.5 cm$^{-1}$, that results in 8461 spectral channels for each single spectrum. Since MetOp platforms are on a polar orbit, IASI works as a cross-track scanner, with 30 effective Fields Of Regard (FOR) per scan, spanning an angle range of $\pm$48.33 degrees on both sides of nadir. Each FOR consists of a 2$\times$2 matrix of so-called Instantaneous Fields Of View (IFOVs); each IFOV has a diameter of 14.65 mrad, corresponding to a ground

resolution of 12 km at nadir at a satellite altitude of 819 km. The IFOVs matrix is centred on the viewing direction. Hence, at nadir, the FOR of the four IASI pixels projects at the ground a square area of $\sim$50$\times$50 km. A more exhaustive description of IASI, the mission objectives and its achievements can be found in (Hilton et al. , 2012). IASI data were downloaded from the EUMETSAT data centre.

For the present study, we delimited a geographic box around the Lampedusa island, defined by the latitude range [35.0° N

36.0° N] and longitude [12.0° E 13.2° E]. Morning IASI soundings were collected in this area in three different days: 22 and 28 June, and 3 July 2013. These three days are characterized by a decreasing atmospheric desert dust load: this is the ideal case to characterize the effectiveness of aerosol parameters in different conditions. The geographic area of interest, together with the exact location of IASI soundings, are reported in Fig. 2. Only clear-sky FOVs are indicated; they are pre-selected through a scene analysis software based on a Cumulative Discriminant Analysis approach (Amato et al. , 2014). A further selection is

made discarding all the spectra affected by sunglint contamination, which are not useful to characterize the accuracy of aerosol spectral properties at wave lengths shorter than 5 $\mu$m.

### 3.2   Radiative transfer input data and methods

On the side of simulations, the radiative transfer model relies on proper input data for surface properties, atmospheric state and aerosol vertical distributions. As far as emissivity is concerned, since all the IASI FOVs that we examine involve sea surface,

emissivity is computed according to Masuda's emissivity model (Masuda et al. , 1988). We have developed a look-up table with sea surface emissivity over the IASI spectral range and resolution. The emissivity has been pre-computed for VZA values spanning the interval from 0° to 50° (at steps of 1°) and for an average wind speed of 5 m/s.

For surface temperature and atmospheric status, we used the analyses provided by the European Center for Medium range Weather Forecasts (ECMWF) collocated with IASI measurements. ECMWF provides data at the four canonical hours (0, 6,

12 and 18 UTC) on a spatial grid mesh of 0.125°$\times$0.125 °. Since we do not retrieve atmospheric and surface parameters, they are estimated as the temporal average between the two consecutive ECMWF analyses closest to IASI soundings (mostly acquired around 9 UTC), namely those at 6 and 12 UTC of the three days that we consider. More specifically, we use ECMWF temperature, water vapour mixing ratio and ozone mixing ratio profiles (T,Q,O), and surface temperature $T_s$. All these profiles are originally provided on either 91 pressure levels (until 25 June 2013) or 137 pressure levels (from 26 June 2013). Since the


ECMWF pressure grid is not fixed, depending on surface pressure, all the atmospheric profiles are preliminarily averaged on the 60-layer pressure grid of $\sigma$-IASI-as. The other gases required as input from the radiative transfer model are tuned according to the most recent updates of climatology (WDCGG , 2015). The final, average (T,Q,O) input profiles used for the IASI pixels effectively exploited in the three selected dates are showed in Fig. 3.

The last input required by radiative transfer is the aerosol vertical profile and dimensional distribution. Such data were acquired from the Forward Scattering Spectrometer Probe (FSSP-300), the Ultra High Sensitivity Aerosol Spectrometer (UHSAS), which are two spectrometers, the two particle counters GRIMM-1 and GRIMM-2 and the Scanning Mobility Particle Sizer (SMPS) on board the ATR-42 aircraft (Denjean et al. , 2016). These instruments measure the particles concentration in different size ranges; for this radiative closure experiment, we use the particles concentration measurements for diameters in the interval

0.05-30 $\mu$m, at different altitudes. The dimensional distributions have been fitted with a variable number (typically 4 or 5) of log-normal modes. Different sets of log-normal modes have been found to be representative of the size distributions observed at different altitudes. The observed particle size distributions are represented for the three selected days in Fig. 4, while the retrieved log-normal modes to reproduce them are summarized in Tab. 2.

This ensemble of input data has been accustomed to perform a double set of radiative transfer simulations: the first one

includes the radiative impact of aerosols, the second one does not (clear sky simulations). The double simulation is aimed to better characterize the radiative impact of aerosol with respect to IASI real observations. The results reported in Denjean et al. (2016) put in evidence that, in the selected dates, the observed atmospheric aerosols have a variable geographic origin: that observed over Lampedusa on 22 June has two distinct origins, according to the altitude, coming from Southern Morocco for altitudes below 1500 m, and from the Southern Algeria for altitudes above 3500 m. On 28 June, instead, the observed aerosol

comes entirely from Tunisia, while for the case of 3 July, the lower layer (below 3000 m) comes from southern Morocco, the upper from Tunisia. All these information is summarized in Tab. 2, and has been properly considered in the choice of refractive indices used for radiative transfer simulations. For each date, two simulations have been performed: the first using the indices borrowed from Shettle and Fenn (1979), and included in the OPAC database, the second working with sets of refractive indices which account for the parent soils of the observed aerosol, that have been initially derived in Di Biagio et al. (2014) and more

extensively in Di Biagio et al. (2016). In the first study, extinction spectra in the 2-16 $\mu$m range have been measured in situ (T=293 K, RH<2%) for poly-dispersed pure dust aerosols generated from natural parent soils from Tunisia, Niger, and the Gobi desert. Such data have been used in combination with particle size distributions to reckon the complex refractive index of each dust sample. All the sets of refractive indices chosen for the present work are plotted in Fig. 5. They are averaged on a 15 cm$^{-1}$ spectral grid, the same used by $\sigma$-IASI-as to compute the aerosol extinction.

It is fair to stress that a more complex simulation should also consider the presence of polluted aerosol particles in the boundary layer, and their mixing with dust particles. However, we have neglected such effects, since IASI radiances (as shown e. g. in Vandenbussche et al. (2013)) have usually a very limited sensitivity to aerosols in the atmospheric layers where mixing takes place; also, in cases like this, where the observed target is sea surface, the thermal contrast between surface and the boundary layer is limited to 2-4 K (see Fig. 3), and cuts down the sensitivity to aerosols in the boundary layer. Moreover, the concentration

of polluting particles is at its seasonal minimum, because of the absence of sources such as biomass burning. As a conclusive





remark about this aspect, in Denjean et al. (2016) it is shown that pollution particles have only a small influence on the optical properties of the dust plumes over the Western Mediterranean, and that other mixing effects, like coating of polluting species on dust particles have no relevant effect on their optical behaviour. Hence, in these conditions, the direct comparison between simulations and IASI observations in the thermal infrared, relying on well-characterized dimensional distributions, can be

a way of validating the goodness of dust indices, and to probe the sensitivity of satellite-based, hyperspectral, infrared Earth observations to the geographical origin of aerosol. We perform a direct comparison between the spectral residuals obtained with the two sets of indices, both spectrum by spectrum, and on the day-by-day average, in order to see whether or not significant, systematic discrepancies occur.

## 4  Results

### 4.1  Single spectra results

Here we first show sample spectra with the aim to give an idea of the accuracy of the model with respect to the single spectrum simulation, and to provide a first, qualitative feedback about the difference between clear-sky calculations, those in presence of dust aerosol, and IASI spectra. Figures 6 and 7 represent two single spectra (one for 22 June and the second acquired on 3 July) among those (see Fig. 2) selected for the analysis. The two plots capture the most salient spectral signatures that affect

IASI radiances; among them, the $\nu_2$ and $\nu_3$ $CO_2$ absorption bands, centred at 667 cm$^{-1}$ and 2385 cm$^{-1}$, the $O_3$ band at 1040 cm$^{-1}$, and the wide $H_2O$ $\nu_2$ absorption band centred at 1590 cm$^{-1}$, and many other minor features in the three atmospheric windows 780-980 cm$^{-1}$, 1070-1200 cm$^{-1}$ and beyond 2440 cm$^{-1}$. A more detailed focus on these regions is given in the same figures, which is important since the aerosol extinction effect is heavily manifested therein, with differences in brightness temperature units that can be as large as 5 K for the case of 22 June. At a first glance, we can see that the radiance computed

with $\sigma$-IASI-as including the dust aerosol impact reproduces fairly the radiance as observed by IASI. The slope due to the behaviour of complex refractive index in the spectral region around 1000 cm$^{-1}$ is well manifested both in the computed and observed radiances. It is proper to point out that both the surface temperature and the water vapour columnar amount provided by ECMWF and used by the model have been slightly tuned in order to better match IASI observations, which is necessary to correct some well-known biases typical of ECMWF re-analyses, and because we do not perform any retrieval of the *true state*

*vector*.

As a general remark, we observe that, in both cases, the simulated radiance is closer to that observed by IASI in the long wave atmospheric windows than in the short wave. Part of this inconsistency is certainly due to the fact that $\sigma$-IASI-as does not reproduce non-LTE effects, which interest especially the $\nu_3$ $CO_2$ absorption band, and can also affect the atmospheric window on that side, but it is likely to assert that this is also due to a poor characterization of refractive indices in the short wave,

because of the continuous shape of the misfit around 2400 cm$^{-1}$.



## 4.2 Residuals analysis

The conclusion expressed above is much more reinforced if we perform a more detailed analysis on spectral residuals on the average of the IASI soundings for each one of the three days we have considered. In this discussion, we examine also in detail the difference between the radiance simulated with the two sets of refractive indices cited before.

Figure 8 shows the whole, average observed and simulated spectra (with and without dust aerosol), a zoom on the spectral windows, and the differences between observed and calculated radiances computed over the 12 selected spectra for the high aerosol load case (22 June). The first three plots seem to confirm what has been already observed on the single spectra, while new insights are revealed by residuals (last panel). Having not retrieved in any way atmospheric and surface parameters from spectra, some of the structures seen in the residuals are those due to biases within ECMWF analyses. Among them, the evident misfit of the $CO_2$ band at 667, 720 and 750 cm$^{-1}$, which has been already identified in Liuzzi et al. (2016), and related to the fact that ECMWF analyses actually tend to overestimate stratopause temperature and to underestimate the tropopause and high troposphere temperature, both by 2-4 K. An average bias of 1 K is also manifested in the spectral region interested by the water vapour absorption, which again is related to issues in ECMWF water vapour profiles. The last major bias, due to radiative transfer limitations, is the large misfit in the $\nu_3$ band of $CO_2$, caused by the already cited negligence of non-LTE effects in the code.

As far as aerosols are concerned, their average effect is to adjust the spectral slope in the atmospheric windows, reducing the difference with the observed IASI average spectrum. Anyway, there are significant discrepancies between the two sets of refractive indices: if we use the indices by Shettle and Fenn , they generate an inconsistency between the atmospheric windows at the two sides of the ozone band, with an evident misfit around 950 cm$^{-1}$, and a continuous residual as large as 1.5 K in the interval 1070-1200 cm$^{-1}$. Such residuals are largely suppressed in the interval 780-980 cm$^{-1}$ by adopting the more recently derived indices by Di Biagio et al. , while they do not solve the issues in the interval 1070-1200 cm$^{-1}$. The same conclusion can be drawn for the short wave spectral window, where the same continuous slope appears using both the refractive index sets. To summarize, in this first case it seems that, the choice of using refractive indices like those by Di Biagio et al. , which account for the geographic source of the aerosol, solves part of the inconsistencies which are observed with, to say, a standard set of indices: in this sense, IASI data reflect the specific provenance of the observed aerosol. Nevertheless, some spurious features in the residuals still remain using both sets, suggesting that further efforts are needed to characterize the specific refractive indices in the thermal infrared.

Similar effects appear in the comparison between observations and simulations for the other two days, which are reported in Fig. 9 and 10. In these two cases, the most abundant aerosol is that coming from Tunisia. In the case of 28 June, we observe that the new indices by Di Biagio et al  suppress the slope introduced using the indices by Shettle and Fenn , which is as large as 1.5 K, and attenuates the misfit in the short wave, while inconsistencies still hold between the two atmospheric windows at the sides of the ozone band. Analogous observations can be done in the case of 3 July, both in the long wave and in the short wave windows. In this last spectral interval, since the same slope (with a larger shift between observed and calculated) is visible in the clear sky residuals, and its magnitude decreases together with the aerosol load, it is likely to argue that such





inconsistency in the short wave is due both to an incorrect input sea emissivity, which depends critically on wind speed on sea surface (Masuda et al. , 1988) especially above 2000 cm$^{-1}$, and to a poor characterization of dust refractive indices in this spectral region.

A quantitative comparison between residuals is better summarized in Table 3. We stress that the standard deviations in Table 3 are computed on spectral channels, and not on the single spectra. To clarify, if $N$ is the number of spectral points (at a resolution of 15 cm$^{-1}$) in the spectral window we are analyzing, and $R(\sigma_i)$, $i = \{1, \ldots, N\}$ the average, spectrally-binned radiance vector, where the average is made over the $M$ spectra we consider, the standard deviation is computed as

$$\left[ \frac{1}{N-1} \sum_{i=1}^{N} \left( R(\sigma_i) - \bar{R} \right)^2 \right]^{\frac{1}{2}}$$

where $\bar{R}$ is the average spectral residual in that particular spectral window. In this way, the standard deviation brings information about the spurious variability introduced by an incorrect characterization of the refractive index, and can be really compared with IASI radiometric noise.

In the short wave spectral window (last column), the two indices exhibit the same standard deviations, which are above the IASI noise (Serio et al. , 2015) only in the case of 22 June. This puts in evidence the need to better characterize dust aerosol spectral properties in this region for aerosols coming from Algeria and Morocco, while the input indices for Tunisian aerosol give satisfactory results. In the long wave, instead, the indices by Di Biagio et al. steadily reduce the average RMSD and standard deviations below the IASI average noise level in all the three cases in the spectral window 780-980 cm$^{-1}$. Instead, in the atmospheric window 1070-1200 cm$^{-1}$, the average residual is better for these indices than those computed using the indices by Shettle and Fenn , while for the cases where the dominant aerosol is that from Tunisia, the new indices yield double average residuals, and generally larger standard deviations. To summarize, in the long wave the refractive indices for the aerosol coming from Algeria and Morocco seems to perform better than those by Shettle and Fenn , which is the opposite of the results obtained in the short wave.

On the side of radiative transfer scheme, taking into account that the short wave slope decreases together with the aerosol load, we can argue that the multiple scattering scheme adopted by $\sigma$-IASI-as is actually effective in reproducing the observed radiance, at least with the aerosol loads we have observed in these cases. In the long wave atmospheric windows, where scattering effects are of second order with respect to pure absorption, the model works within the IASI radiometric noise average level, especially with the more recent indices, and the "absorption approach" demonstrates to be effective.

# 5 Conclusions

In this work we have provided an in-depth description of the new $\sigma$-IASI-as radiative transfer model, and used it to probe the goodness of the characterization of the microphysical properties of desert dust aerosol over Lampedusa in the context of the ChArMEx experiment, in three scenarios with a variable aerosol load.

The radiative transfer model we have presented is based on a pseudo-monochromatic approach as far as gases optical depth



calculation is concerned, while it exploits ab-initio Mie calculations to reckon aerosol extinction. This makes the model extremely feasible with respect to the aerosol properties, as it has been showed throughout the whole paper. Moreover, the model has been not tailored to any particular instrument, thus it is immediately available, in particular, for the next generation of hyper-spectral sensors (e.g. IASI-NG, MTG-IRS) whose operativity is scheduled in the next ten years.

The calculations we have performed have exploited the characterization of dimensional distributions of desert dust particles over Lampedusa performed by dedicated instruments. The high level of detail of these observations has enabled to assess, in particular, the effectiveness of refractive indices used to model desert dust aerosol extinction, by comparing different models. Overall, we find that dust aerosol modelling, in terms of dimensional distributions, reproduces with a fair degree of accuracy the observed radiance in the thermal infrared wave lengths, while presents some troubles in the short wave portion of the IASI

spectrum where, anyway, also other major effects occur (e.g. non-LTE). The discrepancies observed between IASI radiances and calculated spectra in this region evidence the need to better characterize dust extinction in the mid-near infrared (as already stated in other contexts, Carboni et al. (e.g. 2012)), especially in case where the observed aerosol has heterogeneous geographical source. In the thermal infrared, instead, the observed residuals are generally better, and compatible with the uncertainties that affect the other radiative transfer input data (ECMWF atmospheric profiles and surface properties). However, better con-

sistency has to be achieved between different spectral intervals.

This validation study is a first step toward the possibility to retrieve directly from IASI radiances the microphysical properties of clouds and aerosols, namely their vertical distribution and/or particles dimensional distributions, thanks to the fact that the radiative transfer model we have elaborated has the capability to compute analytical or semi-analytical Jacobians of the radiance also with respect to particles concentration.

*Acknowledgements.* IASI has been developed and built under the responsibility of CNES. It is flown onboard the MetOp satellites as part of the EUMETSAT Polar System. The IASI L1 data are received through the EUMETCast near real-time data distribution service. External calibration IASI L1 data and corresponding engineering data are provided by the IASI Technical Expertise Centre at CNES.



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





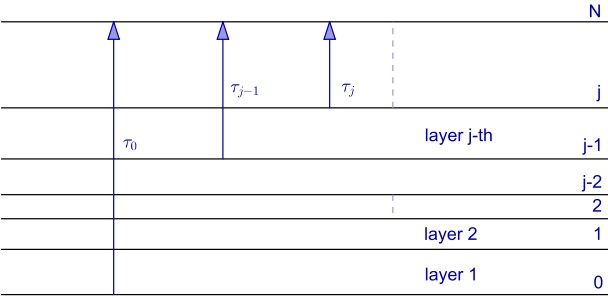

**Figure 1.** Layering scheme of the atmosphere, with the layers numbering and the definition of transmittances.

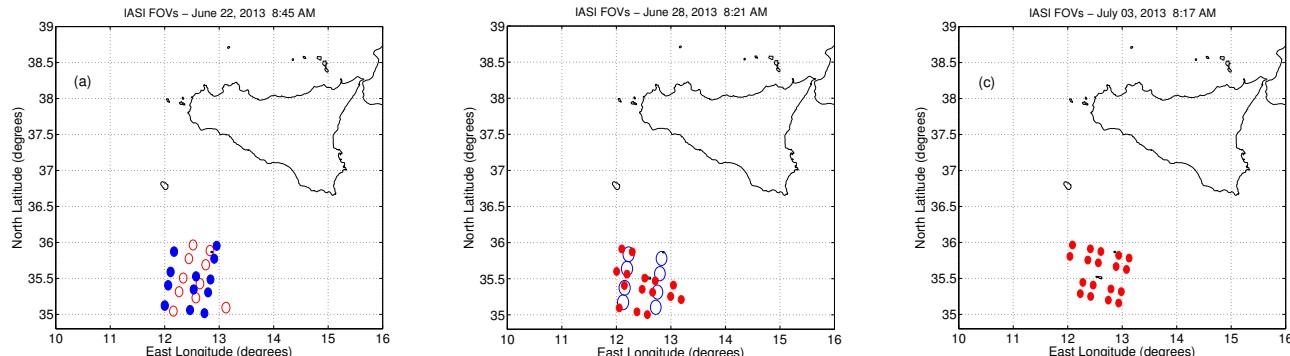

**Figure 2.** Location of the IASI Fields of View analysed in this work. From the left to the right: 22 June, 28 June, and 3 July 2013. Red circles correspond to those acquired from IASI onboard of MetOp-A, those in blue from IASI onboard of MetOp-B. Filled footprints are those that have been effectively used, while those unfilled have not been exploited because of strong sunglint contamination in the short wave.

Warren, S. G.: Optical constants of ice from the ultraviolet to the microwave, Appl. Opt., 23, 1206–1225, doi:10.1364/AO.23.001206, 1984.

WDCGG W, WMO, World Data Centre for Greenhouse Gases (WDCGG) data summary, vol.39, 115 pp., 2015.





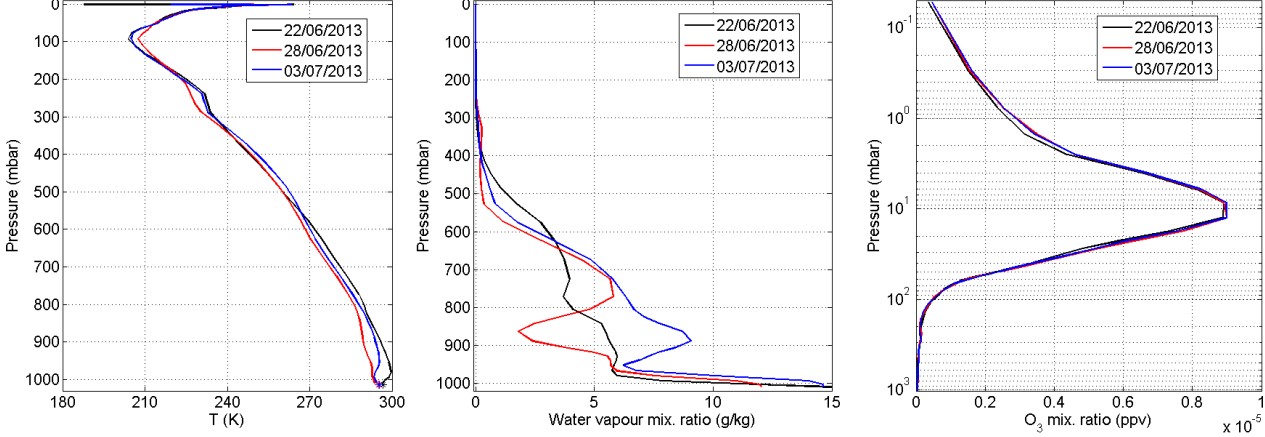

**Figure 3.** From the left to right: ECMWF spatial-temporal average profiles for temperature (left), water vapour (middle) and ozone (right) in the three days involved in this analysis. Together with the temperature profiles, ECMWF average surface temperatures for each day are also indicated by markers.

**Table 1.** List of the aerosols included in the $\sigma$-IASI-as model. [a] sea salt refractive indices are available for eight different values of relative humidity, (0% to 99%). [b] $H_2SO_4$ droplets are available for two temperature values: 215 K and 300 K. [c] birefringent materials: each of them has two sets of refractive indices, one for the ordinary light ray, and the other for the extraordinary one. This is convenient in case $\sigma$-IASI-as is used to simulate polarized radiances.

| Aerosol types | |
|---|---|
| *Water droplets* | *Ice crystals* |
| NaCl | Sea salt[a] |
| Hydrophilic aerosol | $NH_3$ droplets |
| Carbonaceous aerosol | Volcanic dust |
| $H_2SO_4$ droplets[b] | Meteoric dust |
| Quartz[c] | Hematite[c] |
| Desert sand[c] | Saharan dust |
| Volcanic ash | Flame soot |
| Ammonium sulphate | Burning vegetation ash |





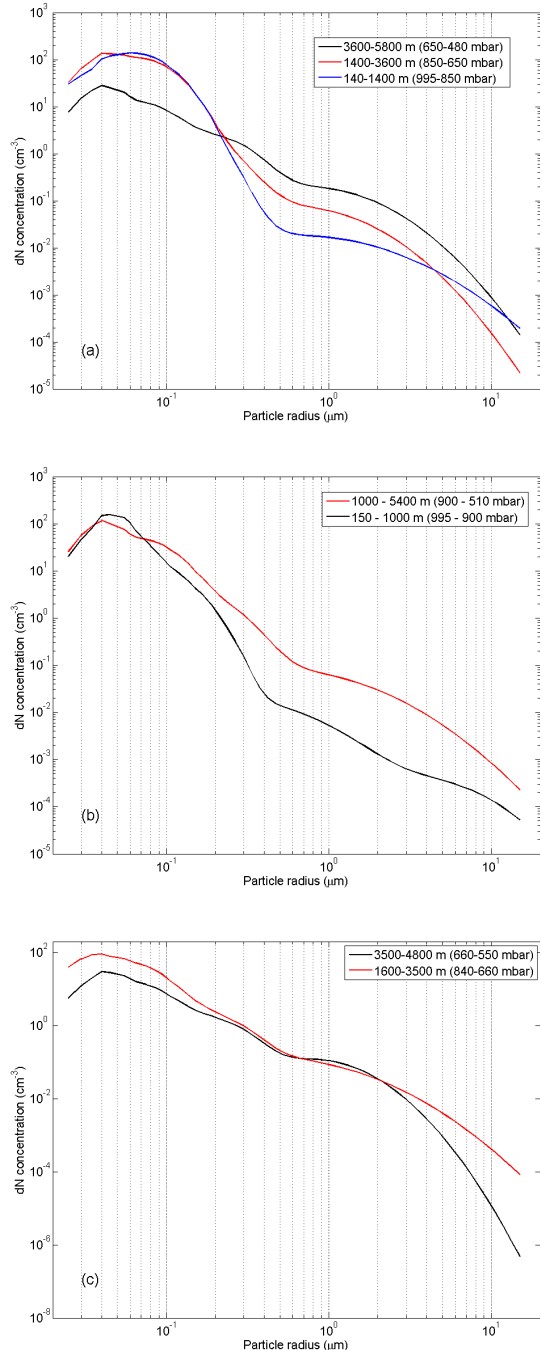

**Figure 4.** Measured dimensional distributions of dust aerosol by the instruments on board the ATR-42 at different altitudes: (a) 22 June; (b) 28 June, and (c) 3 July 2013. In the first case, there are three distinct dust layers, instead of two. The extra layer (blue curve) is that closest to the surface: in that day, dust aerosol is more abundant than in the other two cases.





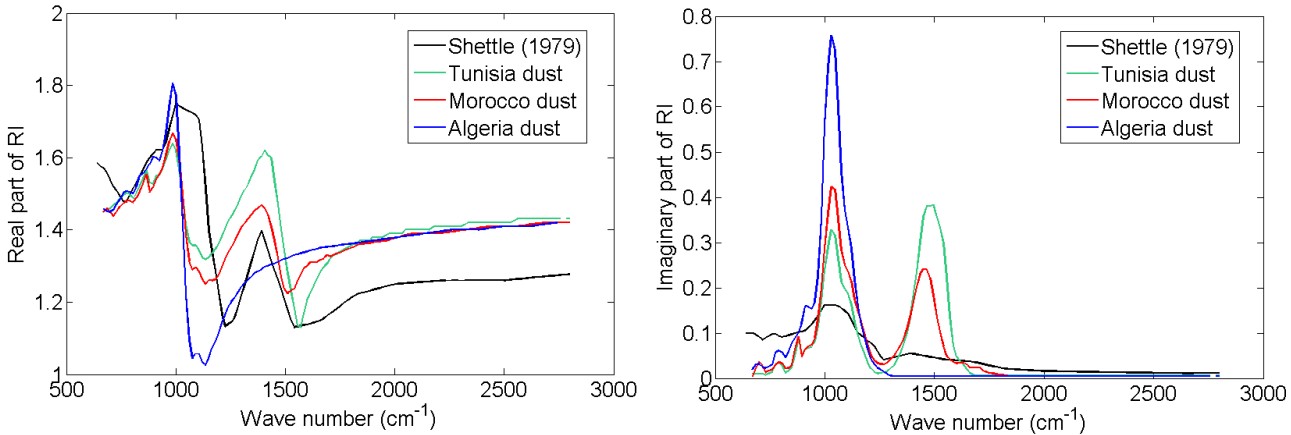

**Figure 5.** Complex refractive indices used as input in the radiative transfer calculations. Left plot: real part of Shettle and Fenn (black curve) and of the three indices (red, green and blue curves) derived by Di Biagio et al. ; right plot: same as left, but for the imaginary part.

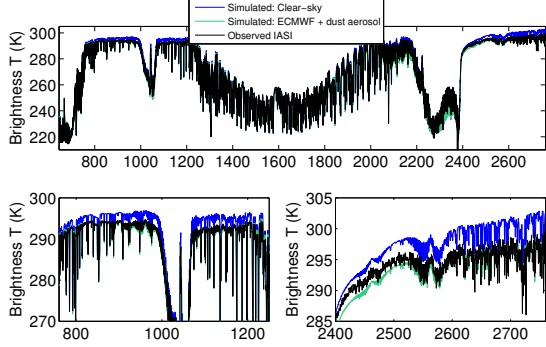

**Figure 6.** Sample IASI spectrum (black) acquired on 22 June (higher dust aerosol load) compared with radiative transfer simulations in absence (blue) and presence (green) of dust aerosol. Refractive indices are those borrowed by the work by Di Biagio et al. .

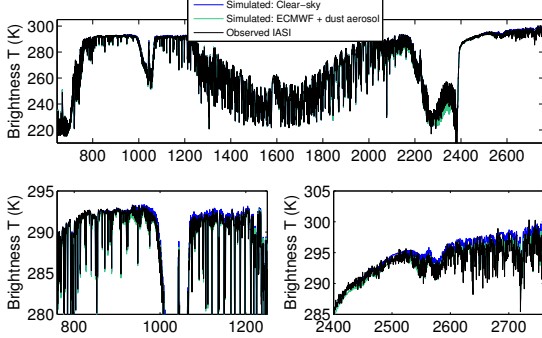

**Figure 7.** Same as Fig. 6, with a second spectrum acquired on 3 July (low aerosol load).



**Figure 8.** Top panel: IASI average spectrum (black) of the 12 soundings acquired on 22 June, compared to the average of the computed spectra: clear sky (blue), with dust aerosol, using Shettle and Fenn refractive indices (red), and Di Biagio et al. indices (green); middle panels: zoom of the spectra in the IASI long wave (left) and short wave (right) atmospheric windows; bottom panel: residuals computed in the three cases.



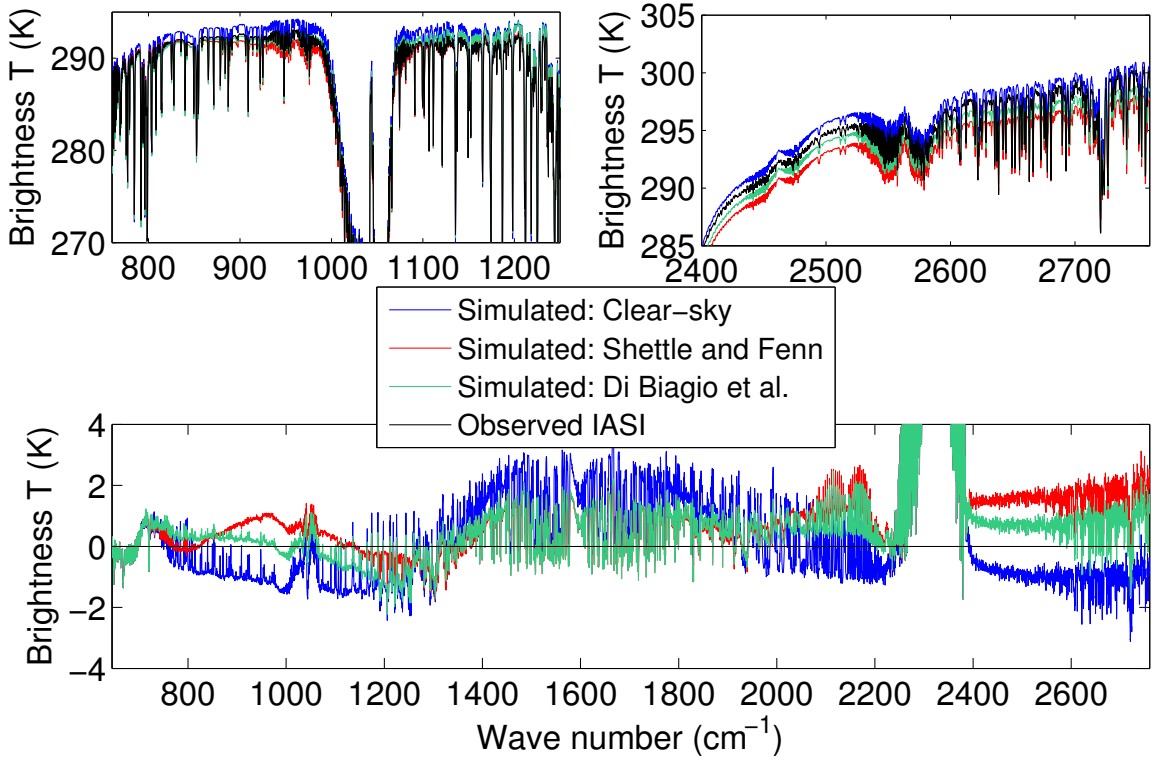

**Figure 9.** Top and bottom panels: same as middle and bottom panels, for the case of 28 June (medium aerosol load). Average residuals are computed over 15 IASI soundings.





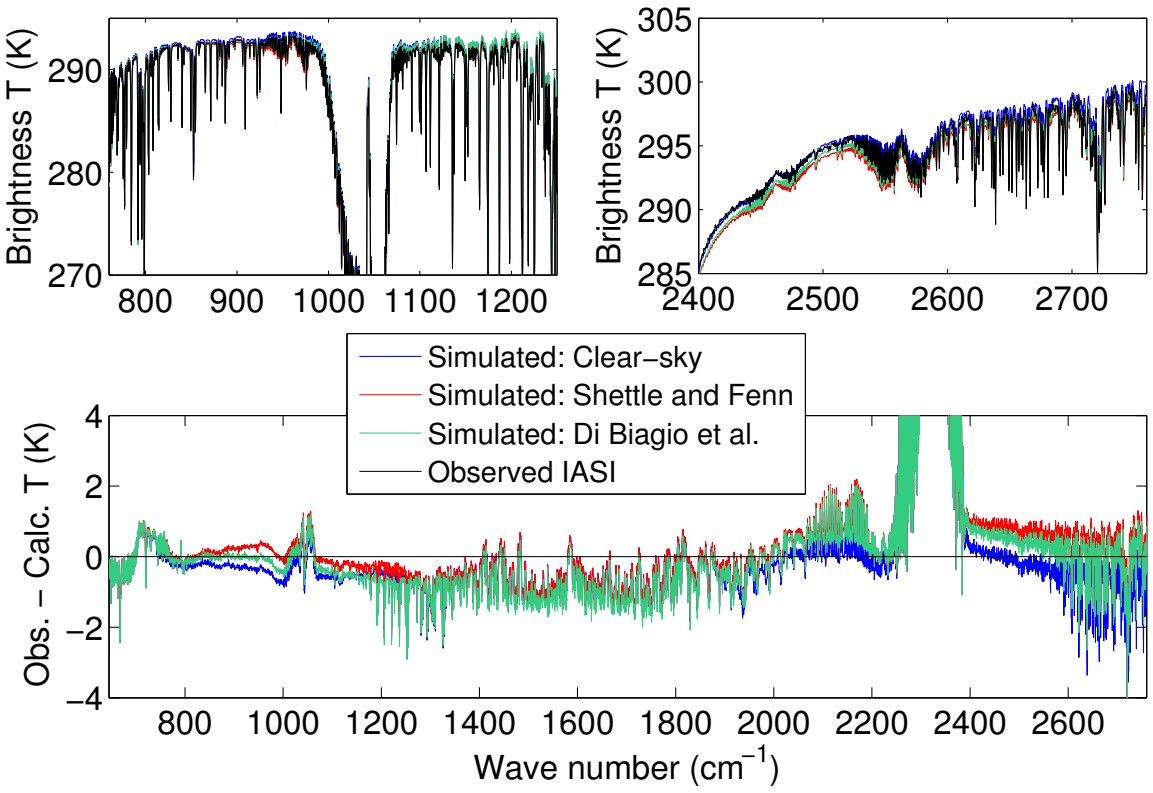

**Figure 10.** Same as Fig. 9, for the case of 3 July (low aerosol load). Average residuals are computed over 18 IASI soundings.



**Table 2.** From the top to the bottom: log-normal modes used to fit the observed size distributions on 22 June (top table), 28 June (middle one) and 3 July (below) for each altitude range (first column).

| | | Mode 1 | Mode 2 | Mode 3 | Mode 4 | Origin |
|---|---|---|---|---|---|---|
| **140-1400 m** | $N_{tot}$ | 500.00 | 900.00 | 5.00 | 0.40 | |
| | $R_0$ ($\mu$m) | 0.045 | 0.070 | 0.140 | 0.675 | Southern Morocco |
| | $\sigma_0$ | 1.50 | 1.50 | 1.55 | 2.80 | |
| **1400-3600 m** | $N_{tot}$ | 500.00 | 900.00 | 8.00 | 1.20 | |
| | $R_0$ ($\mu$m) | 0.039 | 0.070 | 0.185 | 0.650 | Southern Morocco |
| | $\sigma_0$ | 1.28 | 1.50 | 1.55 | 2.18 | |
| **3600-5800 m** | $N_{tot}$ | 130.00 | 100.00 | 20.00 | 3.20 | |
| | $R_0$ ($\mu$m) | 0.039 | 0.070 | 0.200 | 0.775 | Southern Algeria |
| | $\sigma_0$ | 1.28 | 1.50 | 1.55 | 2.18 | |

| | | Mode 1 | Mode 2 | Mode 3 | Mode 4 | Mode 5 | Origin |
|---|---|---|---|---|---|---|---|
| **150-1000 m** | $N_{tot}$ | 800.00 | 200.00 | 10.00 | 0.35 | 0.01 | |
| | $R_0$ ($\mu$m) | 0.045 | 0.065 | 0.150 | 0.250 | 3.940 | Tunisia |
| | $\sigma_0$ | 1.30 | 1.50 | 1.35 | 2.40 | 2.00 | |
| **1000-5400 m** | $N_{tot}$ | 500.00 | 400.00 | 20.00 | 1.70 | | |
| | $R_0$ ($\mu$m) | 0.039 | 0.070 | 0.175 | 0.500 | | Tunisia |
| | $\sigma_0$ | 1.25 | 1.50 | 1.55 | 2.70 | | |

| | | Mode 1 | Mode 2 | Mode 3 | Mode 4 | Origin |
|---|---|---|---|---|---|---|
| **1600-3500 m** | $N_{tot}$ | 300.00 | 500.00 | 20.00 | 2.50 | |
| | $R_0$ ($\mu$m) | 0.035 | 0.055 | 0.165 | 0.450 | Southern Morocco |
| | $\sigma_0$ | 1.25 | 1.50 | 1.55 | 2.50 | |
| **3500-4800 m** | $N_{tot}$ | 100.00 | 160.00 | 15.00 | 2.00 | |
| | $R_0$ ($\mu$m) | 0.040 | 0.060 | 0.185 | 0.800 | Tunisia |
| | $\sigma_0$ | 1.25 | 1.50 | 1.55 | 1.80 | |



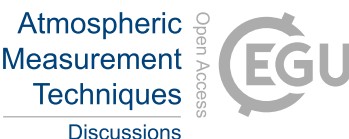

**Table 3.** Summary of root mean square differences and related standard deviations computed using the two sets of refractive indices in the three selected scenarios. Data are computed averaging the residuals on a spectral grid with a sampling of 15 cm$^{-1}$, the same used by $\sigma$-IASI-as to compute aerosol extinction, in order to suppress most of the variability due to gases spectral lines, retaining only the variability due to aerosol extinction.

| | Ref. Index | RMSD $\pm$ Std. dev. (K) | | |
| --- | --- | --- | --- | --- |
| | | 780-980 cm$^{-1}$ | 1070-1200 cm$^{-1}$ | 2440-2760 cm$^{-1}$ |
| 22 June | Shettle and Fenn | 0.134$\pm$0.460 | 0.288$\pm$0.160 | 0.165$\pm$0.760 |
| | Di Biagio et al. | 0.039$\pm$0.130 | 0.161$\pm$0.380 | 0.224$\pm$0.780 |
| 28 June | Shettle and Fenn | 0.175$\pm$0.400 | 0.082$\pm$0.260 | 0.338$\pm$0.160 |
| | Di Biagio et al. | 0.077$\pm$0.080 | 0.225$\pm$0.220 | 0.156$\pm$0.160 |
| 3 July | Shettle and Fenn | 0.057$\pm$0.140 | 0.085$\pm$0.090 | 0.117$\pm$0.380 |
| | Di Biagio et al. | 0.019$\pm$0.055 | 0.185$\pm$0.130 | 0.081$\pm$0.370 |
| **IASI av. noise (K)** | | 0.123 | 0.137 | 0.577 |