# Peer review of "Consistency of dimensional distributions and refractive indices of desert dust measured over Lampedusa with IASI radiances"

_Atmospheric Measurement Techniques, 2016_

## Referee Comment (RC1) · Anonymous Referee #1 · 19 Nov 2016

In this study, the authors make use of airborne measurements made over the Western Mediterranean during the ADRIMED campaign, as well as a dataset of complex dust aerosol refractive indices obtained in the lab to test the capability of the newly developed s-IASI-as radiative transfer code to mimic the IASI spectra in the presence of dust advected from Africa over the sea. The study makes use of 3 days of measurements of ADRIMED, which are not presented (they are extensively covered in Denjean et al. 2016), used to constrain the s-IASI-as algorithm. The strengths and limitations of the s-IASI-as algorithm are discussed on the basis of the comparison with real IASI spectra in the 700-2800 cm-1 spectral range.

Overall, this is an interesting study which deserves publication in AMT. The quality

of the English could be improved. Also, some relevant references are missing which should be included. Finally, the authors should to a more thorough job at quantifying the errors and uncertainties in the different part of the spectrum. Also the impact of these errors and uncertainties on the aerosols microphysical properties (as presented in Section 2.2) derived from s-IASI-as.

Based on the above, I recommended that the paper be published in AMT provide that the above mentioned comments as well as the ones below are taken into account.

Recommendation: minor revision

Abstract - Define ChArMEx - "a dust event which occurred": based on the following, you have not looked at just a single dust event - there is no such thing as the southern med basin: here you are referring to the southern part of the Western Med basin - lines 5-9: the sentence is too long and conveys more than one idea, rephrase

Introduction Page 1 - L15-16: there is also some work on the topic from Lars Kluser at DLR - L21-22: Note that profiling of dust was recently achieved over Eastern Asia by the group of Juan Cuesta in Creteil. Please cite:

J. Cuesta, M. Eremenko, C. Flamant, G. Dufour, B. Laurent, G. Bergametti, M. Höpfner, J. Orphal and D. Zhou, 2015: Satellite observation of the 3D distribution of dust plumes from IASI throughout a major desert dust outbreak across East Asia in March 2008, J. Geophys. Res., 120, 7099-7127, doi:10.1002/2014JD022406

Page 2 - L19: a high degree of generality - L20: the maximum fastness: lame, please rephrase - L27-28: complicated sentence: just write that ADRIMED was a component of ChArMEx SOP1 - L29: is the ATR-42 an aircraft? Who is the operator, SAFIRE?? Please provide some details. - L33: later (p 10) you write that on 22 June some of the dust is coming from Southern Morocco, which is not the Sahara.. please correct.

Data and methods Page 9 - L14-15: could you specify the number of IASI pixels in the box? What is the time of the IASI overpasses on the 3 days? - L16: the 3 days are

characterized by a decreasing atmospheric dust load: unclear, please rephrase. - L18: here and elsewhere: should this be IFOV? - L27: is this wind speed at 10 m above sea surface? How does 10-m wind speed relate to surface sea surface emissivity? In case of strong winds, whitecaps develop at the sea surface that will modify significantly the surface emissivity. It would be worth checking what the sea state and surface wind speeds are around Lampedusa during the 3 days selected.

Conclusions P13 - L30: cases rather than scenarios P14 - L9-10: unclear sentence, please rephrase

---

## Short Comment (SC1) · 13 Dec 2016

I have found this manuscript very interesting and useful for the community. Nevertheless, as the Referee#1 mentioned, in the present manuscript there are several missing references that would have helped to better introduce the work done here and its importance.

One obvious case, very evident to me because it deals with my own research, is in the first sentence: "The discussion about the possibility to detect and characterize atmospheric aerosols in the thermal infrared spectral range is relatively recent (Hollweg et al. , 2006; Clarisse et al. , 2013) and very active...", where I suggest to add the following reference:

Sellitto, P. and Legras, B.: Sensitivity of thermal infrared nadir instruments to the chemical and microphysical properties of UTLS secondary sulfate aerosols, Atmos. Meas. Tech., 9, 115-132, doi:10.5194/amt-9-115-2016, 2016

which, in fact, circumscribes and specialises the cited work of Clarisse et al., 2013 on the specific case of sulfate aerosols. The paper demonstrates that TIR observations contain some information to identify specific chemical compositions (and microphysical properties) of atmospheric aerosols, by studying in details their spectral signatures. This is peculiar of the TIR spectral region.

This is just one case but I strongly suggest to thoroughly check the manuscript to solve the issue of missing references. This would add value to the very interesting work done by the authors.

My best regards.
* * *

---

## Editor Comment (EC1) · X. Querol (Editor) · 13 Dec 2016

Dear Dr. P. Sellitto,

Thanks a lot for your short comment.

Yes, it seems evident that, as clearly stated by one of the referees, 3 points need deffinitively be addressed in the revision:

1. The English usage could be improved. 2. Relevant references are missing (including the one you are pointing out). 3. Quantifying the errors and uncertainties, and evaluating the impact of these errors and uncertainties on the aerosols microphysical properties.

[Figure]

Yours sincerely

Xavier Querol

---

## Referee Comment (RC2) · Anonymous Referee #2 · 20 Dec 2016

Review of the manuscript "*Consistency of dimensional distribution and refractive indices of desert dust measured over Lampedusa with IASI radiances*" by Liuzzi et al.

The paper presents the comparison between observed (IASI) and calculated ($\sigma$-IASI-as) radiances during a dust event affecting Lampedusa Island. The novel $\sigma$-IASI-as radiative transfer model takes into account the effects of aerosols on extinction.

**General comments**:

1) First of all I would encourage the authors to explain better the differences between $\sigma$-IASI and $\sigma$-IASI-as models. If I have well understood the $\sigma$-IASI computes the atmosphere spectral radiance both for clear and cloudy sky and the $\sigma$-IASI-as model includes also the effects on spectral radiance due to atmospheric aerosols. The authors state that the effects of particles and clouds on extinction are based "on the same physics". However, for example, multiple scattering has a major importance in clouds rather than in a dust layer. Some of the equations reported in this manuscript (Liuzzi et al., AMTD) have been already discussed in Amato et al. (2002) however there are some differences which I do not understand. For example equation 1 in the present manuscript corresponds to equation 7 in Amato et al. (2002) but there are two differences in the third right term of the equation: $\tau_0$ is $\tau_0^2$ in Amato et al. (2002) and $\tau_*^f$ is $\frac{1}{\tau}$ in Amato et al. (2002). What are the reasons for these differences? The same in the equations 3 and 4 in Liuzzi et al. AMTD (corresponding to equations 12 and 13 in Amato et al. (2002)). Equation 2 in Liuzzi et al. AMTD is exactly the equation 8 in Amato et al. (2002) but a different nomenclature is used ($R_C$ instead of $R_0$ and $R_N$ instead of $R_{cld}$). Is there any reason for this? This is a bit confusing and the same nomenclature as in Amato et al. (2002) should be used in this manuscript, unless the terms included in the equations of Liuzzi et al., AMTD are different, but this is not clear.

2) Is it possible to know $\omega(\sigma)$, $b(\sigma)$ and $g(\sigma)$ computed by the model and compare these intensive aerosol properties with their experimental determinations reported in literature for dust?

3) Pag. 11, Lines 19-22. It is nice to see that the radiance computed with $\sigma$-IASI-as, which includes the impact of dust aerosols, reproduces quite well the observed IASI radiance compared to the simulated clear-sky radiance. However, the authors state that the slope due to the peak in the complex refractive index around 1000 cm$^{-1}$ "is well manifested both in the computed and observed radiances". However, I cannot clearly see the effect of dust absorption on the simulated radiance in Fig. 6 and 7 because of the absorption due to $O_3$ in the same spectral region. Moreover, I do not see any effect of dust absorption at 1500 cm$^{-1}$ in Figs. 6 and 7. Is this due the fact that the absorption by $H_2O$ is dominant in this spectral region?

4) Not sure if this was already discussed in Di Biagio et al. What is the reason for the lack of peak in the imaginary part of RI for Algeria dust around 1500 cm$^{-1}$?

5) The English must be improved

6) Uncertainties of $\sigma$-IASI-as outputs should be discussed.

---

## Author Comment (AC1) · 17 Jan 2017

REPLY TO REFEREE #1

Referee comments are in highlighted text

Author reply is in normal text

In this study, the authors make use of airborne measurements made over the Western Mediterranean during the ADRIMED campaign, as well as a dataset of complex dust aerosol refractive indices obtained in the lab to test the capability of the newly developed s-IASI-as radiative transfer code to mimic the IASI spectra in the presence of dust advected from Africa over the sea. The study makes use of 3 days of measurements of ADRIMED, which are not presented (they are extensively covered in Denjean et al. 2016), used to constrain the s-IASI-as algorithm. The strengths and limitations of the s-IASI-as algorithm are discussed on the basis of the comparison with real IASI spectra in the 700-2800 cm-1 spectral range. Overall, this is an interesting study which deserves publication in AMT. The quality of the English could be improved. Also, some relevant references are missing which should be included. Finally, the authors should to a more thorough job at quantifying the errors and uncertainties in the different part of the spectrum. Also the impact of these errors and uncertainties on the aerosols microphysical properties (as presented in Section 2.2) derived from s-IASI-as.

We would like to thank the referee for the kind appreciation of the manuscript, whose importance in the context of the existing literature has been fully understood. Also, we thank the referee for the productive comments, which have been very useful to improve the overall quality of the manuscript. However, we point out that, apart from testing the code capabilities, the paper tries to pursue two further objectives: on the one hand, the σ-IASI-as code is fully presented and explained, in order to show the updates to the pre-existing work done in Amato et al. (2002). On the other hand, it has been shown that IASI data are sensitive to the aerosol optical properties, and consequently to their origin. We have somehow reinforced these concepts rephrasing a bit the abstract. As suggested, we have improved the quality of the English and the abundance of proper references.

As far as the last point is concerned, the referee should consider that the manuscript deals simply with radiative transfer simulations in presence of dust aerosol whose dimensional properties are measured, and optical behaviour discussed. We are not doing any retrieval of surface, atmospheric, or aerosol properties, and the uncertainties are those of radiative transfer, which are already discussed in the previous paper by Amato et al. (2002), where it is shown that σ-IASI outputs are affected by errors <<0.1 K in the thermal infrared. This is far lower than IASI radiometric noise. This aspect has been commented in the revised paper at page 5, L8-9 and page 9, L6-8. Another point is that, the code reckons aerosol and clouds optical depth using exact Mie routines, and avoids any parameterization. Single and multiple scattering effects are considered as well, even if in this case one does not expect that multiple scattering is dominant, which is instead the case, e. g., of dense clouds. Likewise, the manuscript is not intended to pursue any comparison between a full treatment of aerosol extinction and our code. Instead, to better discuss IASI data sensitivity with respect to aerosol properties, as requested, we added a new section 4.3, in which the effect of dimensional distributions uncertainty on radiative transfer outputs is discussed.

Based on the above, I recommended that the paper be published in AMT provide that the above mentioned comments as well as the ones below are taken into account.

Recommendation: minor revision

Abstract - Define ChArMEx - "a dust event which occurred": based on the following, you have not looked at just a single dust event - there is no such thing as the southern med basin: here you are referring to the southern part of the Western Med basin - lines 5-9: the sentence is too long and conveys more than one idea, rephrase

We have rephrased the first sentence, rewriting "in presence of atmospheric desert dust, between June and July 2013 in the southern Mediterranean basin, in the air mass above Lampedusa island", in order to avoid confusion about the context of IASI observations. Lines 5-9 have been made more clear rephrasing in this way: "Simulations have been carried on using as input different sets of input complex refractive indices, which take into account the parent soils of the aerosols. Their accuracy also relies on the high-quality characterization of desert dust microphysical properties, achieved through direct measurements in the ChArMEx experiment. On the one hand, the fact that the model can ingest such a variable input proves its feasibility". Since (see previous comment) we have added also a new section in which we analyze the sensitivity of IASI radiances with respect to the effective radius of the particles, we have mentioned it in the revised abstract (L10-11): "and pursues an assessment of the sensitivity of IASI data with respect to the dimensional distribution of desert dust particles".

Introduction Page 1 - L15-16: there is also some work on the topic from Lars Kluser at DLR - L21-22: Note that profiling of dust was recently achieved over Eastern Asia by the group of Juan Cuesta in Creteil. Please cite:

J. Cuesta, M. Eremenko, C. Flamant, G. Dufour, B. Laurent, G. Bergametti, M. Höpfner, J. Orphal and D. Zhou, 2015: Satellite observation of the 3D distribution of dust plumes from IASI throughout a major desert dust outbreak across East Asia in March 2008, J. Geophys. Res., 120, 7099-7127, doi:10.1002/2014JD022406.

We thank the referee for this comment. The following two works by L. Kluser have been cited at page 9, L17-19:

- Klüser, L., Kleiber, P., Holzer-Popp, T., and Grassian, V. H.: Desert dust observation from space - Application of measured mineral component infrared extinction spectra, Atmospheric Environment, 54, 419–427, doi:10.1016/j.atmosenv.2012.02.011, 2012.
- Klüser, L., Banks, J. R., Martynenko, D., Bergemann, C., Brindley, H. E., and Holzer-Popp, T.: Information content of spaceborne hyperspectral infrared observations with respect to mineral dust properties, Remote Sensing of Environment, 156, 294–309, doi:10.1016/j.rse.2014.09.036, 2015.

This has also helped to better present the way in which our manuscript contributes to provide an advancement to the existing research. Finally, the suggested reference by Cuesta et al. has been added at Page 1, L17.

Page 2 - L19: a high degree of generality - L20: the maximum fastness: lame, please rephrase

The referee is right. To avoid confusion, and since most of the significance of this lame sentences is better explained later in the manuscript, we have eliminated this sentences, and added a comment in L28-29: "Also, since the model ingests the optical properties of several aerosol and cloud types, it can be applied to simulate infrared observations including all the variability of such properties."

- L27-28: complicated sentence: just write that ADRIMED was a component of ChArMEx SOP1

- L29: is the ATR-42 an aircraft? Who is the operator, SAFIRE?? Please provide some details.

To answer to both the points, we have rephrased the whole paragraph between Line 32, Page 2, and L3, Page 3: "Thus, for this validation exercise, we have fruitfully employed the aerosol microphysical properties (i.e. dimensional distribution and concentration) derived during the Chemistry-Aerosol Mediterranean Experiment (ChArMEx) Special Observation Period SOP1a. One of the components of SOP1 was the Aerosol Direct Radiative Impact on the regional climate in the MEDiterranean region (ADRIMED) field campaign. Here, we exploit the measurements made on board the ATR-42 aircraft, operated by SAFIRE (Service des Avions Francais Instruments pour la Recherche en Environnement, http://www.safire.fr/) over Lampedusa, in four different days between June and July 2013."

- L33: later (p 10) you write that on 22 June some of the dust is coming from Southern Morocco, which is not the Sahara.. please correct.

To avoid misunderstandings, we have replaced "Saharan" with "North-western African".

Data and methods Page 9 - L14-15: could you specify the number of IASI pixels in the box? What is the time of the IASI overpasses on the 3 days? - L16: the 3 days are characterized by a decreasing atmospheric dust load: unclear, please rephrase.

To answer to both comments, and to better arrange this part of the manuscript, former L16 of the manuscript has been rewritten in this way: " These three days are characterized by different atmospheric desert dust loads and vertical profile shapes", while the former L14-15 have been rewritten at L6-8, Page 10, as follows: "This pre-filtering process yield to a total of 12 IASI IFOVs (out of 22 clear sky) actually used in this work, for the case of 22 June; 15 IFOVs (out of 23) for the 28 June case; finally, 18 IFOVs for the case of 3 July. The acquisition times for the three days are, respectively, 8h45m UTC, 8h21m UTC, and 8h17m UTC."

- L18: here and elsewhere: should this be IFOV?

It has been corrected to IFOV in the "Data and methods" section, since it is referred to the IASI instrument. Instead, in the previous section, we have used simply "FOV", since it is simply intended to be the Field of View of a generic instrument.

- L27: is this wind speed at 10 m above sea surface? How does 10-m wind speed relate to surface sea surface emissivity? In case of strong winds, whitecaps develop at the sea surface that will modify significantly the surface emissivity. It would be worth checking what the sea state and surface wind speeds are around Lampedusa during the 3 days selected.

As stated in the article, we use the Masuda model (Masuda et al., 1988) to describe sea surface emissivity. If the wind speed is in the interval 0-15 m/s, wherein the model holds, emissivity differences are as large as 0.004 in case the Viewing Zenith Angles is the maximum IASI VZA (~48 degrees), or lower for lower VZA values. As a consequence, we expect no significant bias in radiative transfer simulations in case wind speed is lower than 15 m/s. Indeed, we have verified (source: ECMWF reanalyses) that wind speed at 10 m (that to be considered in Masuda model) is 3 m/s, 8.5 m/s, and 4 m/s respectively in the three days we work with. Consequently, the sentence at Page 13, L1-2 has been eliminated, since it was clearly incorrect.

Conclusions P13 - L30: cases rather than scenarios

Corrected. Moreover, since a consistent part of the manuscript is dedicated to the characterization of dust refractive indices, we have mentioned it at L19, page 15.

P14 - L9-10: unclear sentence, please rephrase

We have rephrased as follows: " Overall, we find that the dimensional distributions derived from ChArMEx observations yield to a fair consistency between observations and calculations in the thermal infrared wave lengths. On the contrary, the discrepancies are significant in the short wave portion of the IASI spectrum where, anyway, also other major effects occur (e.g. non-LTE)."

---

## Author Comment (AC2) · 17 Jan 2017

REPLY TO REFEREE #2

Referee comments are in highlighted text

Author reply is in normal text

First, on behalf of all co-authors, the corresponding author would like to thank the referee for the valuable comments provided, that have been very useful to improve the quality of the manuscript.

The paper presents the comparison between observed (IASI) and calculated (σ-IASI-as) radiances during a dust event affecting Lampedusa Island. The novel σ-IASI-as radiative transfer model takes into account the effects of aerosols on extinction.

**General comments:**

1) First of all I would encourage the authors to explain better the differences between σ-IASI and σ-IASI-as models. If I have well understood the σ-IASI computes the atmosphere spectral radiance both for clear and cloudy sky and the σ-IASI-as model includes also the effects on spectral radiance due to atmospheric aerosols.

The differences between the two models have been more extensively explained in Sec. 1 (Introduction, Page 2, L13-19): "The capability of the model to deal with plenty of atmospheric aerosols, with arbitrary vertical profiles and dimensional distributions, is one of the main advancements with respect to the former σ-IASI model (Amato et al. , 2002), which was conceived to work only with simplified single-layer cloud models, in which the cloud was characterized by its own temperature and emissivity. The new version of the model, instead, works with the spectrally variable complex refractive index of aerosols and clouds (water/ice), performing ab-initio Mie calculations, and adopting robust and well-validated schemes for the fast parameterization of multiple scattering effects.". This may help to better introduce the specific explanation of the radiative transfer scheme, in the following Section.

The authors state that the effects of particles and clouds on extinction are based "on the same physics". However, for example, multiple scattering has a major importance in clouds rather than in a dust layer.

The referee is right. Of course, the code considers this difference between clouds and aerosols computing the effect of multiple scattering, as extensively explained. However, in order to avoid any misunderstanding, the sentence has been suppressed.

Some of the equations reported in this manuscript (Liuzzi et al., AMTD) have been already discussed in Amato et al. (2002) however there are some differences which I do not understand. For example equation 1 in the present manuscript corresponds to equation 7 in Amato et al. (2002) but there are two differences in the third right term of the equation: $\tau_0$ is $\tau_0^2$ in Amato et al. (2002) and $\tau^f_*$ is $1/\tau$ in Amato et al. (2002). What are the reasons for these differences? The same in the equations 3 and 4 in Liuzzi et al. AMTD (corresponding to equations 12 and 13 in Amato et al. (2002)).

In the paper by Amato et al. (2002) the reflected term of the radiance was slightly handled (via Equation 5) in order to put in evidence the term $\tau^2_0$, and that this part of the observed radiance is of second order with respect to the atmospheric and surface emitted term. However, the radiative transfer code does not compute the transmittances of the reflected term in the form used in Amato et al., but indeed works in the way explained in the present manuscript. This is the reason why we have used this new notation, which does not yield any substantial difference with respect to the Amato et al. paper. Also, the new notation is useful to better explain the difference between the $\tau^f_*$ term computed in the case of specular reflection or Lambertian diffusion.

Equation 2 in Liuzzi et al. AMTD is exactly the equation 8 in Amato et al. (2002) but a different nomenclature is used ($R_C$ instead of $R_0$ and $R_N$ instead of $R_{cld}$). Is there any reason for this? This is a bit confusing and the same nomenclature as in Amato et al. (2002) should be used in this manuscript, unless the terms included in the equations of Liuzzi et al., AMTD are different, but this is not clear.

Apart from the new physics introduced in the $\sigma$-IASI-as model as far as aerosols and clouds are concerned (see the first review point), the only reason for using $R_C$ instead of $R_0$ is to avoid confusion between the "0" of $\tau_0$ and the "0" of clear sky. Thus, we have replaced the subscript "N" with "cld" where needed, and left unchanged the "C" subscript for clear sky terms.

2) Is it possible to know $\omega(\sigma)$, $b(\sigma)$ and $g(\sigma)$ computed by the model and compare these intensive aerosol properties with their experimental determinations reported in literature for dust?

The way in which these intensive properties are computed is explicitly stated in the manuscript, sec. 2.2, and it is based on Mie routines: their calculation is straightforward, given the dimensional distributions and refractive indices. The radiative transfer code computes all these intensive properties, but we preferred not to report it, given the number of sets of refractive indices and dimensional distributions involved in the manuscript. Instead, we work directly with the observed and computed radiances.

3) Pag. 11, Lines 19-22. It is nice to see that the radiance computed with $\sigma$-IASI-as, which includes the impact of dust aerosols, reproduces quite well the observed IASI radiance compared to the simulated clear-sky radiance. However, the authors state that the slope due to the peak in the complex refractive index around 1000 cm-1 "is well manifested both in the computed and observed radiances". However, I cannot clearly see the effect of dust absorption on the simulated radiance in Fig. 6 and 7 because of the absorption due to $O_3$ in the same spectral region.

The referee is right. The sentence has been modified writing "The attenuation due to the behaviour of complex refractive index at both sides of the $O_3$ band is well manifested both in the computed and observed radiances".

Moreover, I do not see any effect of dust absorption at 1500 cm-1 in Figs. 6 and 7. Is this due the fact that the absorption by $H_2O$ is dominant in this spectral region?

Correct. The water vapour absorption around 1500 cm$^{-1}$ is strong enough to hide the effect of aerosol extinction in the cases we examine, in which the aerosol distribution extends up to an altitude of 5800 m (or lower). To better see this fact, the referee can look at Figure 1 on this report, which shows the derivative of the radiance with respect to desert dust concentration, computed by $\sigma$-IASI-as for the case of 22 June, and averaged on the spectral interval 1400-1600 cm$^{-1}$. In this plot, the derivative is rescaled with dust concentration, so it represents the extinction, in radiance units, due to desert dust in that spectral region, which in this case is negligible (the absolute value is of the order of $10^{-6}$ radiance units).

[Figure]

Figure 1.

4) Not sure if this was already discussed in Di Biagio et al. What is the reason for the lack of peak in the imaginary part of RI for Algeria dust around 1500 cm-1?

This aspect is briefly discussed in the paper by Di Biagio et al, sec. 5.3. The Algerian aerosol is poor in calcite compared to Tunisia and Morocco dust, hence the absorption feature centred at 7 μm, which is typical of calcite, is not present in the refractive index of Algerian dust.

5) The English must be improved.

We have revised the manuscript and rephrased/improved the language where needed.

6) Uncertainties of σ-IASI-as outputs should be discussed.

Here, the referee should consider that the manuscript deals simply with radiative transfer simulations in presence of dust aerosol whose dimensional properties are measured, and optical behaviour discussed. We are not doing any retrieval of surface, atmospheric, or aerosol properties, and the uncertainties are those of radiative transfer, which are already discussed in the previous paper by Amato et al. (2002). There it is shown that σ-IASI uses 60 atmospheric layers to describe the vertical distribution of every parameter (aerosols and clouds included), and that this number of layers produces outputs affected by errors <<0.1 K in the thermal infrared. This aspect has been commented at Page 5, L8-9, and page 9, L6-8. Such uncertainties are far lower than IASI radiometric noise. Moreover, the code reckons aerosol and clouds optical depth using exact Mie routines, and avoids any approximation for scattering effects. Indeed, the manuscript is not intended to pursue any comparison between a full treatment of aerosol extinction and our code. Instead, we have completed the analysis of IASI data sensitivity with respect to aerosol properties by adding a new section 4.3, in which we have better quantified the effect of dimensional distributions uncertainty on radiative transfer outputs. Finally, we have better commented the residuals presented in the manuscript at L2-5, page 13, in order to provide to the reader further elements to better discriminate between those residual features due to desert dust properties and other factors (such as gases or temperature vertical profiles).

---

## Author Comment (AC3) · 17 Jan 2017

Dear Dr. P. Sellitto,

thank you for the interest demonstrated for our paper, and for your short comment. As you suggested, we inserted in the revised manuscript the missing reference you indicated, and others.

Your comment was also very fruitful for us, because it gave us the opportunity to improve the overall explanation of the problem we have addressed, and to give a better form to the results presented in the manuscript.

Sincerely yours

[Figure]

Giuliano Liuzzi (on behalf of all co-authors)